# Effects of Epoxyeicosatrienoic Acid-Enhancing Therapy on the Course of Congestive Heart Failure in Angiotensin II-Dependent Rat Hypertension: From mRNA Analysis towards Functional In Vivo Evaluation

**DOI:** 10.3390/biomedicines9081053

**Published:** 2021-08-20

**Authors:** Petr Kala, Matúš Miklovič, Šárka Jíchová, Petra Škaroupková, Zdeňka Vaňourková, Hana Maxová, Olga Gawrys, Elzbieta Kompanowska-Jezierska, Janusz Sadowski, John D. Imig, John R. Falck, Josef Veselka, Luděk Červenka, Renáta Aiglová, Marek Vícha, Vít Gloger, Miloš Táborský

**Affiliations:** 1Department of Cardiology, University Hospital Motol and 2nd Faculty of Medicine, Charles University, 150 06 Prague, Czech Republic; petrkala@gmail.com (P.K.); josef.veselka@fnmotol.cz (J.V.); 2Center for Experimental Medicine, Institute for Clinical and Experimental Medicine, 140 21 Prague, Czech Republic; mixm@ikem.cz (M.M.); savr@ikem.cz (Š.J.); pesp@ikem.cz (P.Š.); zdva@ikem.cz (Z.V.); olga.gawrys@ikem.cz (O.G.); luce@ikem.cz (L.Č.); 3Department of Pathophysiology, 2nd Faculty of Medicine, Charles University, 150 06 Prague, Czech Republic; hana.maxova@lfmotol.cuni.cz; 4Department of Renal and Body Fluid Physiology, Mossakowski Medical Research Institute, Polish Academy of Sciences, 02-106 Warsaw, Poland; ekompanowska@imdik.pan.pl (E.K.-J.); jsadowski@imdik.pan.pl (J.S.); 5Drug Discovery Center, Medical College of Wisconsin, Wauwatosa, WI 53226, USA; jdimig@mcw.edu; 6Department of Biochemistry, University of Texas Southwestern Medical Center, Dallas, TX 75390, USA; j.falck@utsouthwestern.edu; 7Department of Internal Medicine I, Cardiology, University Hospital Olomouc and Palacký University, I.P. Pavlova 185/6, Nová Ulice, 779 00 Olomouc, Czech Republic; dr.renata.aiglova@seznam.cz (R.A.); dr.marek.vicha@seznam.cz (M.V.); dr.vit.gloger@seznam.cz (V.G.)

**Keywords:** congestive heart failure, volume-overload heart failure, aorto-caval fistula, hypertension, Ren-2 transgenic rats, cytochrome P-450, epoxyeicosatrienoic acids, renin-angiotensin system, angiotensin-converting enzyme inhibitor

## Abstract

This study evaluates the effects of chronic treatment with EET-A, an orally active epoxyeicosatrienoic acid (EETs) analog, on the course of aorto-caval fistula (ACF)-induced heart failure (HF) in Ren-2 transgenic rats (TGR), a model characterized by hypertension and augmented activity of the renin-angiotensin system (RAS). The results were compared with standard pharmacological blockade of the RAS using angiotensin-converting enzyme inhibitor (ACEi). The rationale for employing EET-A as a new treatment approach is based on our findings that apart from increased RAS activity, untreated ACF TGR also shows kidney and left ventricle (LV) tissue deficiency of EETs. Untreated ACF TGR began to die 17 days after creating ACF and were all dead by day 84. The treatment with EET-A alone or ACEi alone improved the survival rate: in 156 days after ACF creation, it was 45.5% and 59.4%, respectively. The combined treatment with EET-A and ACEi appeared to improve the final survival to 71%; however, the difference from either single treatment regimen did not reach significance. Nevertheless, our findings support the notion that targeting the cytochrome P-450-dependent epoxygenase pathway of arachidonic acid metabolism should be considered for the treatment of HF.

## 1. Introduction

Heart failure (HF) has become a major public health problem, affecting currently more than 6.5 and 9.2 million people in the United States of America and the European Union, respectively; the yearly increase in the number of new HF patients is estimated at more than 1.1 million [1,2]. Despite an array of therapeutic approaches available and recent pharmacological advances, the prognosis in HF is still poor, in fact, worse than in common cancers [1,3,4,5,6]. Evidently, new treatment strategies are urgently needed as well as focused experimental studies to evaluate the therapeutic effects of new therapeutic approaches.

Recent research has been focused on the epoxyeicosatrienoic acids (EETs), the metabolites of cytochrome P-450 (CYP)-dependent epoxygenase pathway of arachidonic acid (AA) metabolism. It was shown that EETs importantly contribute to the regulation of renal and cardiovascular function and exert antihypertensive and organ-protective actions [7,8,9,10]. It was also proposed that intrarenal EETs operate as an endogenous compensatory system opposing increased renin-angiotensin system (RAS) activity [7,8,9,11]. Hypertension and inappropriately activated RAS are essential factors promoting the progression of HF [1,2,12,13,14,15,16]; hence, the therapeutic potential of EETs in HF seems promising. EETs are rapidly transformed by soluble epoxide hydrolase (sEH) to biologically inactive dihydroxyeicosatrienoic acids (DHETEs) [7,8,9,17]. Therefore, in most earlier studies, sEH inhibition was employed, and antihypertensive, cardio-, and renoprotective effects were reported [7,8,9]. However, this strategy might prove less successful whenever endogenous EETs biosynthesis is compromised, which might be the case in HF patients receiving the drugs that inhibit CYP activity as an unsolicited action [17,18,19]. Therefore, an alternative approach, which circumvents this limitation, consists of applying EETs-agonistic analogs designed to resist degradation. This new approach has not yet been adequately explored: the obtained results were not entirely consistent and not yet comprehensively evaluated in HF [20,21,22,23,24,25].

In this study, we used Ren-2 transgenic rats (TGR) with HF induced by aorto-caval fistula (ACF). ACF presents a well-defined model of heart failure due to volume overload, characterized by activation of the RAS, congestion, and impairment of renal function. The model has many features in common with untreated human HF [26,27,28,29,30,31,32] and is recommended by the American Heart Association and the European Society of Cardiology for preclinical testing to identify new targets for the treatment of HF patients [33,34]. The TGR model combines activation of the RAS and hypertension [35,36], two well-recognized critical factors for the progression of HF [1,2,6,12,13,16]. We have shown that ACF TGR exhibit markedly enhanced HF-related mortality compared with that in ACF Hannover Sprague-Dawley rats (HanSD), i.e., transgene-negative normotensive controls [8,30,37].

Given the advantages of the described experimental models and availability of 14,15-EETs analog [disodium (S)-2-(13-(3-pentyl)ureido)-tridec-8(Z)-enamido)succinate, EET-A], which was previously found to be suitable for long-term in vivo studies [21,38,39,40], we first aimed to examine effects of chronic EET-A treatment on the morbidity and mortality in ACF TGR and compare it with the standard pharmacological blockade of the RAS with angiotensin-converting enzyme (ACE) inhibitor (ACEi), as described earlier [30,37,41].

In HF patients, the prognosis is worsened when the disease is accompanied by kidney dysfunction (“cardiorenal syndrome”) [3,12,42,43,44]. Therefore, to gain a better insight into the possible role of interactions of CYP-derived eicosanoids with other vasoactive/neurohormonal systems in the pathophysiology of ACF-induced HF, kidney messenger ribonucleic acid (mRNA) expression analysis was performed, with a particular focus on the genes that were previously implicated in the pathophysiology of HF [12]. In addition, to explore in more detail the interactions of CYP-derived eicosanoids and RAS in the pathophysiology of ACF-induced HF, the concentrations of EETs, DHETEs, angiotensin II (ANG II), and angiotensin-1-7 (ANG 1-7) were measured. Moreover, since inappropriate activation of the sympathetic nervous system (SNS) is known to contribute to the progression of HF [14,45,46], the concentrations of norepinephrine (NE) were also measured.

ω-hydroxylase, another CYP-450-dependent enzyme of AA metabolism, generates hydroxyeicosatetraenoic acids (HETEs), mainly 20-HETE [17,47]. Since it might have some role in the progression of HF [48,49], we measured tissue 20-HETE concentrations in this study along with tissue protein expression of CYP2C23 and CYP2J3, the enzymes responsible for EETs formation, and CYP4A1, the enzyme responsible for HETEs production [50]. Also measured was sEH, the enzyme which degrades EETs [7,8,9,10,17]. To obtain knowledge about the neurohormonal activity levels before initiating the treatment regimens, all the parameters mentioned above were assessed in sham-operated TGR, HanSD rats, and untreated ACF TGR two weeks after the ACF operation.

Moreover, to further elucidate the mechanism(s) underlying possible beneficial actions of EET-A on the course of ACF-induced HF, we assessed cardiac structure and function using echocardiography and invasive pressure-volume analysis of the left ventricle (LV). In addition, the renal clearance studies were performed in separate groups of animals. This was done after two weeks of treatment because at this stage untreated ACF TGR began systematically to die. Additionally, in another group of animals that survived until the end, we performed long-term observations (after 20 weeks) and the analyses analogous with those performed within the short-term protocol.

## 2. Materials and Methods

### 2.1. Animals

All animals used in the present study were bred at the Center for Experimental Medicine of this Institute from stock animals supplied by the Max Delbrück Center for Molecular Medicine, Berlin, Germany. Heterozygous TGR were generated by breeding male homozygous TGR with female homozygous HanSD rats as described in the original study [35], age-matched HanSD rats served as transgene-negative normotensive controls. The animals were kept on a 12-h/12-h light/dark cycle. Throughout the experiments, rats were fed a normal salt, normal protein diet (0.45% NaCl, 19–21% protein) manufactured by SEMED (Prague, Czech Republic) and had free access to tap water.

### 2.2. Heart Failure Model, Exclusion Criteria

Eight-week-old male TGR rats were anesthetized with an intraperitoneal injection of ketamine/midazolam mixture (Calypsol, Gedeon Richter, Hungary, 160 mg/kg and Dormicum, Roche, France, 160 mg/kg). Chronic HF due to volume overload was induced by creating ACF using a needle technique. This procedure is routinely performed in our laboratory, and the technique’s details were reported previously [26,31,32,37,51]. Sham-operated rats underwent an identical procedure but without creating the ACF. Animals in which a technical error occurred during ACF creation or a pulsatile flow in the inferior vena cava could not be confirmed, suggesting flawed ACF function, were excluded from the study.

### 2.3. Detailed Experimental Design

#### 2.3.1. Series 1: Assessment of Kidney mRNA Expression in the Early Phase of ACF-Induced HF

The aim was to evaluate the activation of the RAS, CYP-dependent pathways, and the SNS in the kidney at the mRNA level. The animals underwent sham-operation or ACF creation as described above and 14 days later were killed by decapitation. Kidney tissue samples were immediately harvested into liquid nitrogen and stored at −80 °C until analysis. The following experimental groups were examined (n = 9 in each group):Sham-operated HanSD ratsSham-operated TGRACF TGR

The gene expression was determined as described earlier [52]; the procedure is in everyday use in our laboratory [53]. In all experiments, relative gene expression was calculated by the 2−∆∆Ct method, which is most frequently used for such experiments [52]. This method directly uses the Ct (threshold cycle) information generated from a qPCR system. To calculate relative gene expression in target and reference samples, we employed the 18S rRNA gene as a housekeeping gene and used it as the normalizer because its expression level remains relatively stable in response to any treatment [52,53].

First, we calculated ΔCt of each sample following the formula:ΔCt = Ct (gene of interest) − Ct (housekeeping gene)

The expression of mRNA of selected genes was related to that in a control group, i.e., sham-operated HanSD rats. The final results were expressed as the n-fold difference in gene expression of mRNA of target genes between the appropriate experimental group and control group calculated as follows:n − fold expression = 2 − (ΔCt of the experimental group − ΔCt of the control group)

Subsequently, the log transformation of the data was performed to make it more symmetrical, as recommended and generally accepted for evaluation of relative gene expression results [52,53,54,55]. Thus, the values in the graphs represent log2 n-fold gene expression. The measurement of multiple mRNA expressions was performed in accordance with the manufacturer’s instructions (384-well microfluidics TaqMan array cards; custom setting of selected genes; Applied Biosystems, Foster City, CA, USA). The following investigating genes are presented below including the appropriate ID assay identification number and abbreviation given by the manufacturer. The following genes were analyzed in the kidney cortex (Table 1):

#### 2.3.2. Series 2: Assessment of RAS, SNS and CYP Metabolites and Enzymes in the Early Phase of ACF-Induced HF

The aim was to evaluate the degree of systemic (i.e., plasma) and intrarenal activation of two axes of the RAS and SNS. The second aim was to assess the degree of activation of CYP-450-dependent epoxygenase and ω-hydroxylase pathways in the kidney and LV tissues.

Animals were exposed to the same protocol as described in series 1, and the same three experimental groups were evaluated (n = 12 in each group).

Since it is now well recognized that ANG II and ANG 1-7 concentrations in anesthetized animals are higher than those measured in decapitated conscious rats and that normotensive animals exhibit a greater increase in renin secretion in response to anesthesia than observed for ANG II-dependent hypertensive internal renin-depleted animals [36,55,56], at the end of experiments plasma and tissue samples were obtained without anesthesia, i.e., from rats that were killed by decapitation. ANG II and ANG 1-7 were determined by radioimmunoassay, and NE levels were measured by solid-phase enzyme-linked immunosorbent assay. The extracts for EETs, DHETEs, and 20-HETE were separated by reverse-phase high-performance liquid chromatography and analyzed by negative-mode electrospray ionization and tandem mass spectroscopy. In addition, Western blot analysis of protein expression of CYP4A1, CYP2C23, and CYP2J3 in the kidney cortex and LV tissue was performed. These methods are described in detail in our previous studies and are routinely employed in our laboratory [31,32,36,37,38,39,40,53,56,57].

#### 2.3.3. Series 3 and 6: Effects of Treatment with EET-A or ACEi, Alone or Combined, on the Survival Rate and Albuminuria, and Effects of 20-Weeks’ Treatment on Basal Cardiac Function Assessed by Echocardiography and by Pressure-Volume Analysis

Male rats of the same age as in series 1 and 2 derived from numerous litters were randomly assigned to experimental procedures (sham-operation or ACF creation). The animals from a single litter did not prevail in any group. Animals underwent either sham-operation or ACF creation described above on the day labeled as -14 and were left without treatment for 14 days. On a day marked 0, animals that underwent ACF creation were randomly assigned to the specific experimental group. At this time point (day 0), the rats were assigned into the following experimental groups:Sham-operated HanSD rats (initial n = 20)Sham-operated TGR (initial n = 21)ACF TGR + water (i.e., untreated) (initial n = 32)ACF TGR + EET-A (initial n = 32)ACF TGR + ACEi (initial n = 32)ACF TGR + EET-A + ACEi (initial n = 33)

The follow-up period was until day +140 (20 weeks). On the days labeled 0, +14, +28, +70, and +140, the animals were placed in metabolic cages, and 24-h urinary albuminuria was assessed as described previously [32]. To obtain reliable data regarding the effects of treatment regimens on the survival rate and to have a sufficient number of rats to analyze cardiac function and structure by echocardiography and LV pressure-volume analysis, relatively high initial n values were used (also in sham-operated animals). To define such required initial n values, statistical power analysis by the method developed by Cohen [58] was used. In the end, in the survived animals, the cardiac function by echocardiography and subsequently by LV pressure-analysis were assessed as described in detail in our previous studies [37,41,57,59]. In sham-operated animals, twelve animals were randomly assigned for these evaluations.

#### 2.3.4. Series 4: Effects of 2-Weeks’ Treatment with EET-A and ACEi, Alone or Combined, on Basal Cardiac Function Assessed by Echocardiography and by Pressure-Volume Analyses

Animals were prepared as in the previous series. On day 0, the pharmacological treatment was initiated and pursued for 14 days; then, the animals were anesthetized using sodium thiopental (50 mg/kg body weight), and echocardiography and pressure-volume analysis were performed. The following experimental groups (n = 14 in each) were evaluated
Sham-operated HanSD ratsSham-operated TGRACF TGR + waterACF TGR + EET-AACF TGR + ACEiACF TGR + EET-A + ACEi

#### 2.3.5. Series 5: Effects of 2-Weeks’ Treatment with EET-A and ACEi, Alone or Combined, on Renal Hemodynamics and Excretory Function

In this series, the following groups, subjected to the same protocol as in Series 4, were studied (n = 12 in each group):Sham-operated HanSD ratsSham-operated TGRACF TGR + waterACF TGR + EET-AACF TGR + ACEiACF TGR + EET-A + ACEi

At the end of the experiment (day +14), rats were anesthetized using sodium thiopental (50 mg/kg body weight), and acute clearance experiments were performed as described in detail in our previous studies to determine renal hemodynamics and excretory parameters [37,41,59,60].

### 2.4. Statement of Ethics

The study followed the guidelines and practices established by the Animal Care and Use Committee of IKEM, which accord with the national law and were approved by the Animal Care and Use Committee of the IKEM (March 2016) and, consequently, by the Ministry of Health of the Czech Republic (project decision 17124/2016-OZV-30.0-8.3.16/2).

### 2.5. Statistical Analysis

Statistical analysis of the data was performed using Graph-Pad Prism 7 (Graph Pad Software, San Diego, CA, USA). Comparison of survival curves was performed by the log-rank (Mantel–Cox) test followed by the Gehan–Breslow–Wilcoxon test. Statistical comparisons of all other results were made by one-way ANOVA. The values are expressed as the mean ± SEM; *p* < 0.05, indicating a statistically significant difference.

## 3. Results

### 3.1. Series 1 and 2: Assessment of Kidney mRNA Expression and RAS, SNS and CYP Active Agents, Metabolites and Enzymes in the Early Phase of ACF-Induced HF in the Early Phase of ACF-Induced HF

Results of these Series are summarized in Figure 1 and Appendix A. We did not observe significant differences in the kidney mRNA expression of analyzed enzymes and receptors in the early phase of ACF-induced HF. Only the expression of renin in the kidney was significantly reduced in sham-operated TGR and ACF TGR when compared to sham-operated HanSD (Appendix A).

In sham-operated TGR, higher plasma and kidney ANG II levels were observed as compared with sham-operated HanSD rats. At the same time, the ANG 1-7 levels were on the same level in sham-operated TGR as in control HanSD rats, which resulted in the impaired systemic and intrarenal balance between vasodilator and vasoconstrictor axes of the RAS expressed as the ratio of ANG 1-7 to ANG II values (Appendix A). In our previous studies [56,57], this ratio was validated as a reliable marker of the activity of the vasodilator axis of the RAS when the vasoconstrictor axis is hyperactive. Significantly lower values of this index were now observed in plasma and kidneys of sham-operated TGR compared with the sham-operated HanSD rats. Moreover, the creation of ACF significantly increased this index; in the kidney tissue, it reached the levels observed in sham-operated HanSD rats, and in the plasma, it was even higher (Appendix A).

With regards to the activation of the SNS, our results show that there were no significant differences in plasma and kidney NE levels among experimental groups (Appendix A).

Furthermore, there were no significant differences in the concentration of 20-HETE and the expression of CYP4A1 protein in the kidney and LV tissues, which suggests that there were no differences in the CYP-450-dependent ω-hydroxylase pathway of AA metabolism among all experimental groups (Appendix A). Meanwhile, the deficiency in the CYP-dependent epoxygenase pathway of AA was observed in ACF TGR as compared with sham-operated TGR (Appendix A and Figure 1). The kidney and LV tissue deficiency in the biologically active epoxygenase products was revealed mainly in the 11,12-EET and 14,15-EET and it was accompanied by a significant rise of biologically inactive metabolites of CYP-dependent epoxygenase pathway of AA (DHETEs).

Decreased availability of biologically active epoxygenase products expressed as the EETs/DHETEs ratio is commonly used to assess the bioavailability of these metabolites [7,8,9] as was repeatedly validated also in our previous studies [37,38,39,40,41]. As summarized in Figure 1, this ratio was reduced by around 65% in the kidney and LV in ACF TGR as compared with sham-operated TGR. Moreover, as shown in Appendix A there were no significant differences in the kidney and LV protein expression of the enzymes responsible for EETs production among experimental groups. The expression of sEH protein, an enzyme responsible for the conversion of EETs to DHETEs, was significantly increased by the creation of ACF in LV tissue (Appendix A).

### 3.2. Series 3: Effects of Single Treatment with EET-A or ACEi or with a Combination Thereof on the Survival Rate and Albuminuria

All sham-operated HanSD rats and TGR (no ACF) survived until the end of the experiment (omitted for clarity). As shown in Figure 2, untreated ACF TGR began to die on day +3 (17 days after the creation of ACF), and by day +70, all the animals were dead. The treatment with EET-A and ACEi, alone or combined, markedly improved the survival rate, and the final rates were 45.5% (EET-A alone), 59.4% (ACEi alone), and 71% (EET-A and ACEi combined). However, the between-group differences were not significant (TGR ACF + EET-A vs. TGR ACF + ACEi, *p* = 0.25; TGR ACF + EET-A vs. TGR ACF + EET-A + ACEi, *p* = 0.20; TGR ACF + ACEi vs. TGR ACF + EET-A + ACEi, *p* = 0.83; untreated TGR ACF vs. all treatment regimes, *p* < 0.05; sham-operated TGR vs. all treatment regimens *p* < 0.05).

Figure 3 shows that at the start (14 days after sham-operation or creation of ACF, before initiation of treatment), the sham-operated TGR displayed about 20-fold higher albuminuria than the sham-operated HanSD rats. Interestingly, the creation of ACF at this stage caused a significant (about 2.5-fold) decrease in albuminuria. In sham-operated HanSD rats, albuminuria significantly increased throughout the experiment, parallel with the animals’ age. Nevertheless, in the end, it was still significantly lower than in sham-operated TGR at the start of the experiment (3.9 ± 0.2 vs. 8.2 ± 1.6 mg/24 h, *p* < 0.05). In sham-operated TGR, albuminuria was modestly elevated till week 10, but at the end of the study (week 20), it was markedly above the level seen before the start of the experiment (45.1 ± 4.1 vs. 8.2 ± 1.6 mg/24 h, *p* < 0.05).

All treatment regimens further reduced albuminuria in ACF TGR, but the treatments with ACEi, alone or combined with EET-A, were more effective than EET-A alone. In fact, the values of albuminuria at the end of the study were even lower than in the sham-operated HanSD rats at the start of the study (0.14 ± 0.02 and 0.13 ± 0.02 vs. 3.9 ± 0.2 mg/24 h, respectively; *p* < 0.05 in both cases, Figure 3).

The progressive increase in albuminuria in normotensive rats is a natural phenomenon of aging and age-related end-organ damage and in hypertensive rats, it is a generally acknowledged marker of hypertension-related end-organ damage. We have seen this age-dependent progression in sham-operated normotensive, i.e., healthy animals, and in two models of hypertensive rats in our recent studies [32,40]. Nevertheless, it is important to recognize that in sham-operated HanSD rats even if the albuminuria increased with the age, is still minimal. However, the unexpected finding is that two weeks after the creation of ACF the albuminuria was markedly reduced in ACF TGR. This is an entirely surprising finding, especially regarding our recent study in Fawn-hooded hypertensive rats (FHH), a genetic model of spontaneous hypertension associated with chronic kidney diseases (CKD). In this model of high-output HF, the induction of ACF resulted in a progressive rise in albuminuria. In addition, the gradual increase in albumin excretion after the induction of ACF was also found in the control strain, i.e., in Fawn-hooded low-pressure rats (FHL), that do not spontaneously develop CKD [32]. Our working hypothesis is that the decrease in albuminuria in TGR in response to ACF could be mediated by more pronounced initial renal hypoperfusion in ACF TGR as compared with ACF FHH. Fawn-hooded rats, in contrast to TGR, are only slightly hypertensive (systolic blood pressure around 145–150 mmHg), whereas systolic blood pressure in conscious TGR at this age is already 180–190 mmHg. Therefore, the conceivable decrease in renal perfusion pressure in ACF TGR could elicit puzzling beneficial effects on albuminuria. Nevertheless, it is only an assumption and further studies are necessary to address this surprising finding.

### 3.3. Series 4: Effects of 2-Weeks’ Treatment with EET-A or ACEi, Alone or Combined, on Basal Cardiac Function Assessed by Echocardiography and by Pressure-Volume Analysis

Table 2 presents the body weight and organ weight parameters, factored by tibia length. The necessity of such correction is due to significant differences in body weight between untreated ACF TGR and sham-operated TGR, and the rationale for the factoring was documented in our previous work [61,62]. Also shown are the data for all ACF TGR groups exposed to various treatments, displaying remarkable if not significant body weight differences.

The data confirmed that sham-operated TGR displayed whole and LV cardiac hypertrophy as compared with sham-operated HanSD rats. Untreated ACF TGR demonstrated marked bilateral cardiac hypertrophy as seen from the normalized values of the whole heart weight, LV weight (with septum), and the right ventricle (RV) weight, as compared with the sham-operated TGR (no ACF). Moreover, untreated ACF TGR showed significantly higher lung weight than sham-operated TGR, suggesting substantial lung congestion. The treatment with EET-A alone did not alter any of the weight parameters in ACF TGR. In contrast, ACEi alone or combined with EET-A applied in ACF TGR significantly reduced, to a similar degree, the whole heart weight, LV and RV weights, and lung weight.

Table 3 summarizes the evaluation of cardiac function by echocardiography. There were no significant differences in the LV and RV diameters between sham-operated HanSD rats and sham-operated TGR. Nor were there significant differences in the LV and RV function (based on the LV ejection fraction, LV fractional shortening, RV fractional area change, and cardiac output) between sham-operated HanSD rats and sham-operated TGR. Notably, this evaluation corroborates that sham-operated TGR show increased LV anterior and posterior wall thickness and LV relative wall thickness as compared with sham-operated HanSD rats. Untreated ACF TGR exhibited increased stroke volume and cardiac output (the consequence of the shunt), markedly increased LV and RV diameters and decreased relative LV wall thickness (index of the development of eccentric hypertrophy). Also observed was significant impairment of the LV systolic function, as indicated by decreased LV fractional shortening and LV ejection fraction. This occurred without impairment of RV function, as seen from normal RV fractional area change, similar sham-operated TGR. The treatment with ACEi alone or EET-A alone did not change any of the parameters in ACF TGR, but the combined ACEi and EET-A treatment significantly reduced LV anterior and posterior wall thickness as compared with the untreated ACF TGR.

Figure 4 and Figure 5 summarize the evaluation of cardiac function by the invasive hemodynamics method. Sham-operated TGR displayed higher LV peak pressure and maximum rates of pressure rise (+dP/dt)max (Figure 4A,D) and lower maximum rates of pressure fall (−dP/dt)max and of relaxation constant tau (Figure 4E,F) as compared with sham-operated HanSD rats, but there were no significant differences in LV end-diastolic pressure and volume between sham-operated animal groups (Figure 4B,C). Untreated ACF TGR displayed significant decreases in LV peak pressure, (+dP/dt)max and (-dP/dt)max, as compared with sham-operated TGR. None of the treatment regimens attenuated these decreases (Figure 4A,D,E). In addition, untreated ACF TGR exhibited marked increases in LV end-diastolic pressure, LV end-diastolic volume, and LV relaxation constant tau, as compared with the sham-operated TGR, and all of the applied treatments (EET-A and ACEi, alone or combined) significantly attenuated these changes (Figure 4B,C,F).

Sham-operated TGR showed significantly elevated end-systolic pressure-volume relationship (ESPVR) and total peripheral resistance (Figure 5A,D), but also significantly lower LV wall stress (Figure 5F). Untreated ACF TGR displayed substantially decreased ESPVR, the preload recruitable stroke work (PRSW), and total peripheral resistance (TPR) (Figure 5A–D). On the other hand, they showed markedly increased total power output and LV wall stress (Figure 5E,F). The treatment with EET-A and ACEi, alone or combined, did not significantly change any of the parameters in ACF TGR.

### 3.4. Series 5: Effects of 2-Weeks’ Treatment with EET-A or ACEi, Alone or Combined, on Renal Hemodynamics and Excretory Function

Untreated ACF TGR displayed lower mean arterial pressure (MAP) than sham-operated TGR, comparable with sham-operated HanSD rats (Figure 6A). The treatment with EET-A alone or ACEi alone or in combination significantly lowered MAP in ACF TGR, to a similar extent in each of the three groups. There were no significant between-group differences in the glomerular filtration rate (GFR) among all experimental groups (Figure 6B). Untreated ACF TGR exhibited renal blood flow (RBF) significantly lower than in sham-operated TGR (almost 40% reduction). The treatment with EET-A alone or ACEi alone or in combination did not significantly change RBF in ACF TGR (Figure 4C). Untreated ACF TGR displayed substantially lower urine flow and absolute and fractional sodium excretion than sham-operated TGR. None of the treatments normalized these parameters in ACF TGR (Figure 6E,F).

### 3.5. Series 6: Effects of 20-Weeks’ Treatment with EET-A or ACEi, Alone or Combined, on Basal Cardiac Function Assessed by Echocardiography and by Pressure-Volume Analyses

Table 4 summarizes body and organ weights in animals that survived until the end of the study (long-term observation). The degree of the whole and LV cardiac hypertrophy in the ACF TGR treated with ACEi alone was similar as observed in untreated ACF TGR in the early phase (i.e., four weeks after the creation of ACF). Moreover, the RV cardiac hypertrophy was even more pronounced in ACF TGR treated with ACEi at the end of the experiment when compared with untreated ACF TGR in the early phase (see Table 2). Likewise, the ACF TGR treated with EET-A alone showed a similar pattern in cardiac hypertrophy. These results indicate that the cardiac hypertrophy progressed in the animals that were treated with ACEi alone or EET-A alone in the long-term observation. Noteworthy is that the combined treatment with ACEi and EET-A significantly reduced the bilateral cardiac hypertrophy and lung congestion in ACF TGR when compared with solitary ACEi or EET-A.

Table 5 presents the evaluation of cardiac function in the animals that survived until the end of the experiment. All the ACF TGR treated groups showed markedly increased LV diameters that were even higher than those measured in the untreated ACF TGR four weeks after creating ACF. In contrast to the early phase, all ACF TGR groups displayed significant impairment of RV systolic function as indicated by decreased RV fractional area change.

Figure 7 and Figure 8 summarize an evaluation of the cardiac function by the invasive hemodynamics method in animals that survived until the end of the study. At the late stage, the LV end-diastolic pressure in ACF TGR treated with ACEi or EET-A was similar as found in untreated ACF TGR in the early phase (compare Figure 4B and Figure 7B), but the combined treatment with EET-A and ACEi significantly reduced it to the level observed in sham-operated animals (Figure 7B). Moreover, in all the ACF TGR that survived until the end of the experiment, the LV end-diastolic volume was even higher than observed in untreated ACF TGR in the early phase (Figure 4C and Figure 7C), and none of the treatment regimens reduced it (the combined treatment tended to decrease it, NS).

All ACF TGR groups showed decreased ESPVR, EDPVR, PRSW, and TPR compared with the sham-operated TGR (Figure 8A–D). ACF TGR treated with ACEi or EET-A displayed increased LV wall stress compared with the sham-operated TGR, but the combined treatment with EET-A and ACEi significantly reduced it to the level that was not significantly different from that in the sham-operated TGR (Figure 8F).

## 4. Discussion

### 4.1. Activity of the Neurohormonal Systems in ACF-Induced Heart Failure

The degree of activation of the vasoactive/neurohormonal systems was evaluated in the very early phase of the ACF-induced HF model (two weeks after the ACF creation, i.e., just before the treatment regimens were initiated). Our data indicate that in this ANG II-dependent model of hypertension, the creation of ACF leads to the activation of the intrarenal and systemic vasodilatory/natriuretic axis of the RAS (based on ANG 1-7 concentrations and ANG 1-7/ANG II ratio). There were no significant differences in plasma and intrarenal NE concentrations, suggesting no evident activation of the systemic and intrarenal SNS two weeks after the creation of ACF. Thus, the tentative conclusion is that at this very early phase, the major neurohormonal systems, i.e., vasoconstrictor axis of the RAS and SNS were not activated, in contrast to their long-lasting activation and deleterious actions, which was previously described as neurohormonal theory on the progression of HF [14,15,63,64,65]; however, the counter-regulatory axis of the RAS is markedly activated in ACF TGR.

Given the recent discovery that elevated ANG 1-7/ANG II ratio predicts a beneficial outcome of HF [66], our present findings are significant in that they suggest that the initial compensatory response in the ACF-induced model of HF in TGR consists of the activation of the vasodilatory/natriuretic axis of the RAS, which may override the effects of activation of the vasoconstrictor/sodium retaining axis. At first glance, these findings seem to diverge from our previous studies in ACF TGR [37,51,67], in which we observed that plasma and kidney ANG II, aldosterone, and NE levels are much higher than in their sham-operated counterparts. Thus, in this HF model, the vasoconstrictor/sodium retaining axis of the RAS along with the sympathorenal axis appeared to be activated. The reason for this discrepancy might be that our earlier studies were performed five weeks after the creation of ACF [37]. This was done by analogy with the protocol used in normotensive rats, which after five weeks, develop an early, compensated phase HF [28,59,60,68]. However, such selection of the study time proved to be inappropriate in TGR: we demonstrated that the course of high-output HF in TGR is dramatically accelerated [31,37,41,51] when compared both with normotensive and even with spontaneously hypertensive rats (SHR) [69]. In TGR, five weeks after the creation of the ACF, already 50% mortality is observed, compared to simultaneous 100% survival rate in normotensive rats and SHR. Therefore, in the present study, the treatment was initiated earlier (two weeks after ACF), at the stage when 100% survival was still observed. Moreover, our present biochemical data regarding the RAS and SNS show that ACF TGR were indeed in HF’s early compensation phase.

Interestingly, kidney mRNA expression of the rat renin gene was markedly suppressed in sham-operated as well as ACF TGR, which is in accordance with the earlier TGR studies [35,70,71]. Furthermore, it was reported that murine Ren-2 mRNA is expressed at very high levels, especially in the adrenal glands [70,71]. The suppression of the kidney native rat renin gene is mediated through a well-known negative feedback inhibition [72]. Thus, our present findings confirm the consistency of our methods, which is particularly important with mRNA analyses in ACF TGR.

In addition, even though there were no significant between-group differences in kidney mRNA expression of enzymes responsible for the generation and degradation of EETs, our tissue protein expression and biochemical data unequivocally show that kidney and LV tissue availability of EETs in ACF TGR are reduced. This is not the consequence of decreased EETs formation (unaltered protein expressions of CYP2C23 and CYP2J3), but rather the result of increased conversion of EETs to DHETEs, as indicated by increased tissue sEH protein expression in these animals. These findings further support our view that the regulation and activity of neurohormonal/vasoactive systems is a very complex issue and any solid conclusions cannot be made based on the sole mRNA and/or protein analysis. The complexity of these systems requires a comprehensive analysis at all levels and at least measurements of biologically active peptides of the appropriate system.

Furthermore, our results show that renal and LV activity of the CYP-450-dependent ω-hydroxylase pathway of AA metabolism is not increased in ACF TGR, based on unaltered CYP4A1 mRNA and protein expression and 20-HETE tissue concentration. These findings are important because it was proposed that under certain conditions, increased 20-HETE levels can importantly contribute to pathological cardiac hypertrophy, cardiac remodeling, and consequently to the progression of HF [48,49]. Our present findings in ACF TGR do not support this hypothesis. Thus, the second tentative conclusion is that in the early phase, ACF TGR have marked tissue deficiency of biologically active EETs. It has been noticed that EETs importantly protect the myocardium against deleterious cardiac remodeling [73], which is the main factor in the pathogenesis of HF [6,74]. Moreover, they exhibit important renoprotective actions [7,8,9]. Considering our recent findings that ACF TGR display reduced renal vasodilatory responses to EETs [67,75], all of the abovementioned factors support the rationale for treating ACF TGR with EET-A. Furthermore, our data show the dominant role of the tissue deficiency of the 14,15-EETs, in agreement with the accepted knowledge that this isomer is crucial among CYP-dependent epoxygenase metabolites [10,17,19,50]. All this strengthens the basis for employing EET-A in the treatment of ACF TGR.

### 4.2. Effects of the Treatment Regimens on the Survival Rate

We found that the treatment with EET-A alone and ACEi alone considerably attenuated the dramatically aggravated HF-related mortality in ACF TGR. This accords well with our original hypothesis (supported by our comprehensive analyses of CYP-derived metabolites in kidney and LV tissue) that pharmacological enhancement of EETs actions should attenuate HF in ACF TGR, which might have substantial implications for the treatment of HF. Likewise, the beneficial effects of ACEi treatment on the survival rate accord with our line of reasoning: it is generally agreed that even if activation of the vasoconstrictor/sodium retaining axis of the RAS may be initially beneficial, the long-lasting actions might be detrimental and importantly contribute to the progression of HF to the fatal end [6,14,15,63,64]. In addition, in previous studies, we clearly demonstrated that the activity of circulating and intrarenal vasoconstrictor/sodium retaining axis of the RAS is markedly increased in ACF TGR in the advanced HF phase [37,51], and that blockade of this axis by ACEi dramatically improves survival rate [30,37,41].

Collectively, the results demonstrate that both treatments effectively delay the HF-related mortality in ACF TGR, even though each regimen affects specifically only one of the two vasoactive systems. This observation encouraged us to propose that the combined treatment with EET-A and ACEi should enhance the beneficial effects. However, this has not been corroborated: the protection against HF-related mortality was not improved.

Nevertheless, it is emphasized that the final survival rate in the group exposed to the combined treatment with EET-A and ACEi was numerically higher than in groups treated with either drug alone, though, admittedly, the difference did not reach statistical significance. Moreover, the combined treatment displayed some beneficial actions on cardiac morphology and function when evaluated after 20-weeks’ treatment, such as more distinct effects, on the whole, LV and RV cardiac hypertrophy and lung congestion, and also attenuated RV dilatation, in agreement with the data on the organ weight and echocardiography. Furthermore, the pressure-volume analysis of cardiac function at this stage revealed that the combined treatment significantly reduced LV end-diastolic pressure and LV wall stress compared to the treatment with EET-A or ACEi alone. Evidently, with prolonged therapy, the additive beneficial actions on the course of HF-related mortality became apparent. Nevertheless, comprehensive long-term studies with a follow-up period of at least 60 weeks are necessary to draw definite conclusions. Moreover, for reliable analysis of the difference between the relevant survival curves, the initial n values should be at least 42 per group [42,43]. Obviously, such thorough studies are needed even though appropriate prolonged experiments would be difficult and time- and cost-consuming.

### 4.3. Mechanisms of the Beneficial Actions of the Treatments

A search for the mechanism(s) underlying the beneficial actions of each of the treatment regimens on HF-related mortality in ACF TGR was carried out after 2-weeks’ treatment (i.e., four weeks after the creation of ACF) because at this time point untreated ACF TGR were showing strikingly high mortality (only 41% were alive, see the data of series 3). Each treated group showed an almost complete survival rate (85%, 97% and 94%, respectively), which suggested that four weeks post-ACF was the onset point of the decompensation phase of HF in untreated ACF TGR. If so, the beneficial actions of the treatment regimens should be the most effective at this stage.

While the clinical term “cardiorenal syndrome” is a simplification as it does not encompass a spectrum of disorders involving the heart and the kidney [12], there is no doubt that the development of renal dysfunction predicts poor outcomes in patients with HF [3,42,43,44]. We showed previously that ACF TGR seems to be an optimal model for evaluating cardiorenal interaction in the pathophysiology of HF [31,37,51,67]. Moreover, it was revealed that some beneficial impact on the course of high-output HF in ACF TGR can be achieved by preventing renal dysfunction, e.g., by pharmacological blockade of sEH or AT1 receptors [37,51]. However, it was reported that when the pharmacological blockade of RAS by ACE inhibitor (ACEi) was employed in ACF TGR, the protective effect was dominantly mediated by attenuation of cardiac hypertrophy [51]. On the other hand, some authors reported that the development of cardiac hypertrophy after ACF induction is resistant to ACEi treatment [76]; however, this conclusion was based on the studies with normotensive animals with initially normal RAS activity. In addition, the response to the pharmacological blockade of RAS regarding the development of cardiac hypertrophy was reported to be dependent on the phase of cardiac hypertrophy after ACF induction [77]. Considering all these aspects, we have decided to evaluate renal as well as cardiac function at this critical phase to find out if the beneficial actions are achieved predominantly by renal or by cardiac mechanism(s) or by combination thereof.

We found that untreated ACF TGR displayed marked impairment of renal hemodynamics and renal excretory function, in agreement with previous studies in this model [31,37,41,51,67], further supporting the notion that renal dysfunction importantly contributes to the reduced long-term survival rate in this high-output HF model. However, in contrast to our previous study [37], we did not find beneficial effects of ACEi treatment, alone or combined with EET-A, on RBF and renal excretion. Nor were such effects seen with EET-A alone. This was unexpected and somehow discouraging. However, as already discussed above, it is emphasized that beneficial actions of ACEi were seen in the study where the treatment was commenced five weeks after the creation of ACF [37] and lasted for additional five weeks, so it means the beneficial effects of ACEi on renal function were observed in animals that were actually in the very late phase of HF, principally ten weeks after induction of ACF. Thus, the effect of the treatment was explored in untreated ACF TGR in the very late phase of HF. As mentioned in our previous study [37], the major limitation of such protocol is that untreated ACF TGR exhibit almost 50% mortality at this time point, therefore the assessments were performed in an extremely selected population of ACF TGR, which was probably the most resistant subpopulation of animals to the development of high-output HF. To avoid this drawback, our present study was also performed in the early phase of HF. This explanation should reconcile our present and previous findings concerning the effect of treatments with ACEi either alone or in combination with EET-A on renal function in ACF TGR. However, the present finding is unexpected because it indicates that renal mechanisms are not dominantly responsible for the beneficial effects of all the treatment regimens in this study. In this context, it is worth reminding our recent study in which renal denervation improved the survival rate in ACF TGR; however, the beneficial effect was not associated with any improvement of reduced RBF [31]. This is puzzling because it is believed that removing the deleterious influence of renal sympathetic nerve activity, which is the hallmark of HF [14,15,45,46], should improve impaired RBF. This further underscores the complexity of the role of renal dysfunction in the pathophysiology of HF.

Concerning cardiac morphology and cardiac function, we have found that the creation of ACF in TGR resulted after four weeks in marked eccentric chamber remodeling and cardiac hypertrophy, related to the enhanced cardiac output dependent on blood recirculation through the fistula. Load-dependent and load-independent parameters of LV contractility [impaired +(dP/dt)max, suppressed ESPVR, lower PRSW, as well as decreased LV ejection fraction and LV fractional shortening] indicate distinct systolic dysfunction in this very early phase of HF. In addition, the decreased −(dP/dt)max and increased LV relaxation constant indicate impairment of diastolic function in untreated ACF TGR. Simultaneously, untreated ACF TGR exhibited a marked increase in LV end-diastolic pressure and volume accompanied by a pronounced increase in the lung but not liver weight. Taken together, these findings indicate that four weeks after ACF creation, the untreated ACF TGR displayed marked systolic and diastolic dysfunction with signs of developed LV failure, but not RV failure, corresponding to the transition stage from the compensated to the decompensated HF. Each of the treatment regimens attenuated the increase in LV end-diastolic volume, LV end-diastolic pressure, and LV relaxation constant tau (a measure of preload-independent isovolumic relaxation). On the other hand, the systolic and diastolic function parameters, including total power output and LV wall stress, were not favorably changed and did not come close to the values observed in sham-operated TGR. Moreover, the treatment with ACEi, alone or combined with EET-A, significantly attenuated lung congestion in ACF TGR. In contrast, after EET-A alone, the changes did not reach the statistical significance level. However, it is noteworthy that the combined treatment with EET-A and ACEi tended to reduce lung congestion, especially in ACF TGR groups. Taken together, our present findings indicate that all the treatment regimens exhibited some favorable actions on the diastolic function of the LV and attenuated the development of lung congestion in ACF TGR. This improvement was likely responsible for the reduction of HF-related mortality. This would be in agreement with our recent findings that in ACF TGR renal denervation markedly attenuated bilateral cardiac hypertrophy and lung congestion, leading to the conclusion that RDN might be responsible for the improvement of long-term survival [31]. Nevertheless, it is important to acknowledge that our data obtained by echocardiographic analysis in animals that survived until the end of the study after comparing to the results obtained in the very early phase of HF (see Table 2 and Table 4) clearly show that treatment with ACEi, alone or combined with EET-A did not prevent the progression of development of dilated cardiomyopathy, which is one of the hallmarks in this model of high-out HF [28,29,33,34].

Our present and previous results give rise to a crucial question regarding the actual mechanism(s) responsible for the beneficial effects of the treatments applied on LV end-diastolic pressure and lung congestion. Such favorable results were also present at the end of the experiment in the ACF TGR treated with the combination of EET-A and ACEi. Remarkably, at this critical stage, the treatment did not further reduce TPR. Nor did it further decrease TPR at the end of the study compared with groups treated with EET-A or ACEi alone. Therefore, the observed beneficial actions cannot be simply ascribed to reduced afterload. This line of reasoning accords well with our recent findings showing that RDN in ACF TGR exhibited similar beneficial effects but did not alter MAP [31]. Admittedly, we cannot draw decisive conclusions regarding the mechanism(s) responsible for the beneficial actions of the treatments on the LV end-diastolic pressure, LV end-diastolic volume, LV isovolumic relaxation, and, consequently, on lung congestion in the critical phase of HF (transition from compensation to the decompensation phase) in the ACF TGR; further studies are needed to elucidate this issue.

## 5. Conclusions

Despite the apparent limitations of this study, our data show that the ACF TGR in the early phase of high-output HF exhibits substantial tissue deficiency of EETs. This might importantly contribute to developing “cardiorenal syndrome” and progression of HF in this model to the fatal end. There is no doubt that EET-A treatment delays the onset of decompensation of HF and improves the survival rate in ACF TGR; hence, we believe that it could be a promising novel therapeutic approach for the treatment of HF. Admittedly, after the addition of EET-A to ACEi treatment, the survival of ACF TGR only tended to improve compared with the effects of EET-A or ACEi given alone.

Our findings indicate that although ACF TGR develops severe renal dysfunction, the protective effects of either EET-A or ACEi treatments are not dominantly achieved by renal mechanism(s). More likely, beneficial actions on long-term survival are mediated by improving cardiac function, i.e., by reducing bilateral cardiac hypertrophy and lung congestion. This was particularly pronounced after the combined treatment with EET-A and ACEi; however, the specific underlying mechanism(s) remain to be elucidated. Whatever the exact mechanism(s), we believe that our present data strongly support the notion that targeting the CYP-dependent epoxygenase pathway of AA should be considered in attempts to develop new pharmacological strategies for HF treatment.

## Figures and Tables

**Figure 1 biomedicines-09-01053-f001:**
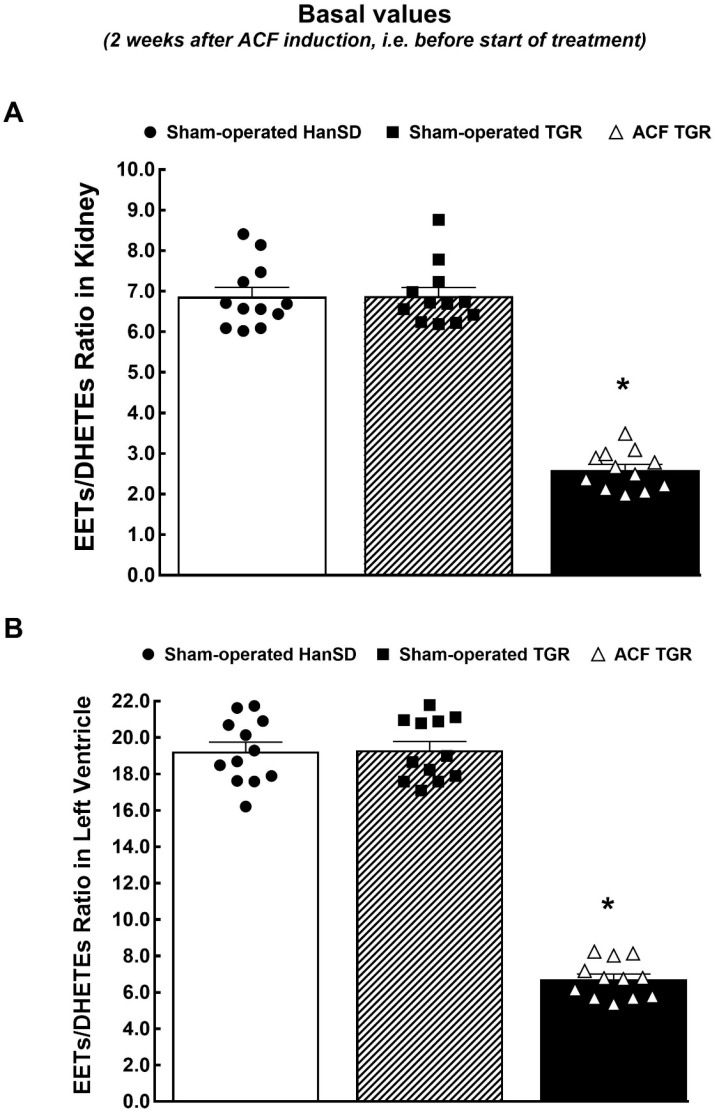
Assessment of kidney and left ventricle tissue availability of biologically active products of cytochrome P-450-dependent epoxygenase pathway of arachidonic acid metabolism. Kidney (**A**) and left ventricle (**B**) ratio in epoxyeicosatrienoic acids (EETs) to dihydroxyeicosatrienoic acids (DHETEs) in sham-operated transgene-negative Hannover Sprague-Dawley rats (HanSD), sham-operated heterozygous Ren-2 transgenic rats (TGR), and TGR rats with aorto-caval fistula (ACF) two weeks after creation of ACF or sham-operation. * *p* < 0.05 versus sham-operated HanSD rats and sham-operated TGR.

**Figure 2 biomedicines-09-01053-f002:**
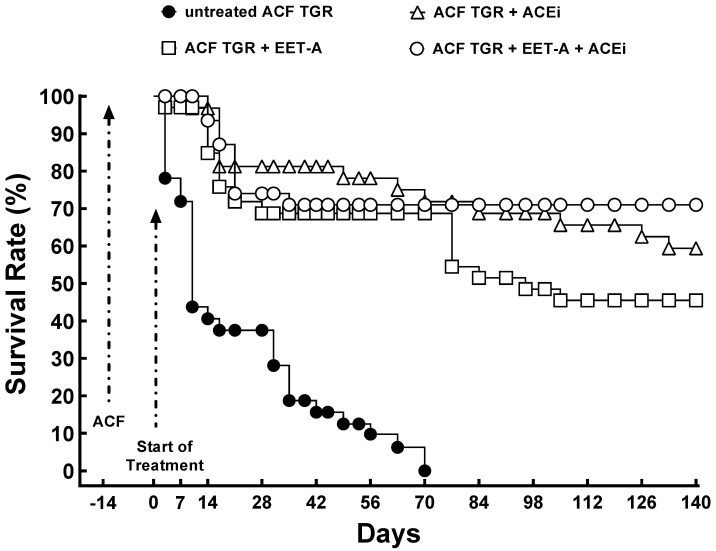
Assessment of the effects of treatment on the survival rate. The survival rate in sham-operated transgene-negative Hannover Sprague-Dawley (HanSD), sham-operated heterozygous Ren-2 transgenic rats (TGR), and TGR rats with aorto-caval fistula (ACF) two weeks after the creation of ACF or sham-operation, treated either with angiotensin-converting enzyme inhibitor (ACEi) alone or with 14,15-epoxyeicosatrienoic acid analog (EET-A) alone or with the combination of EET-A and ACEi.

**Figure 3 biomedicines-09-01053-f003:**
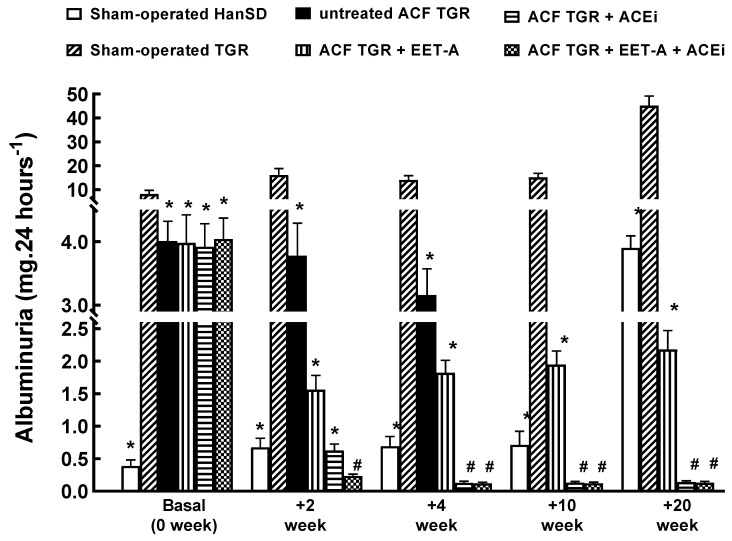
Assessment of the effects of treatment on albuminuria. Albuminuria in sham-operated transgene-negative Hannover Sprague-Dawley (HanSD), sham-operated heterozygous Ren-2 transgenic rats (TGR), and TGR rats with aorto-caval fistula (ACF) two weeks after the creation of ACF or sham-operation, treated either with angiotensin-converting enzyme inhibitor (ACEi) alone or with 14,15-epoxyeicosatrienoic acid analog (EET-A) alone or with the combination of EET-A and ACEi. * *p* < 0.05 versus sham-operated TGR. **^#^** *p* < 0.05 versus sham-operated HanSD rats.

**Figure 4 biomedicines-09-01053-f004:**
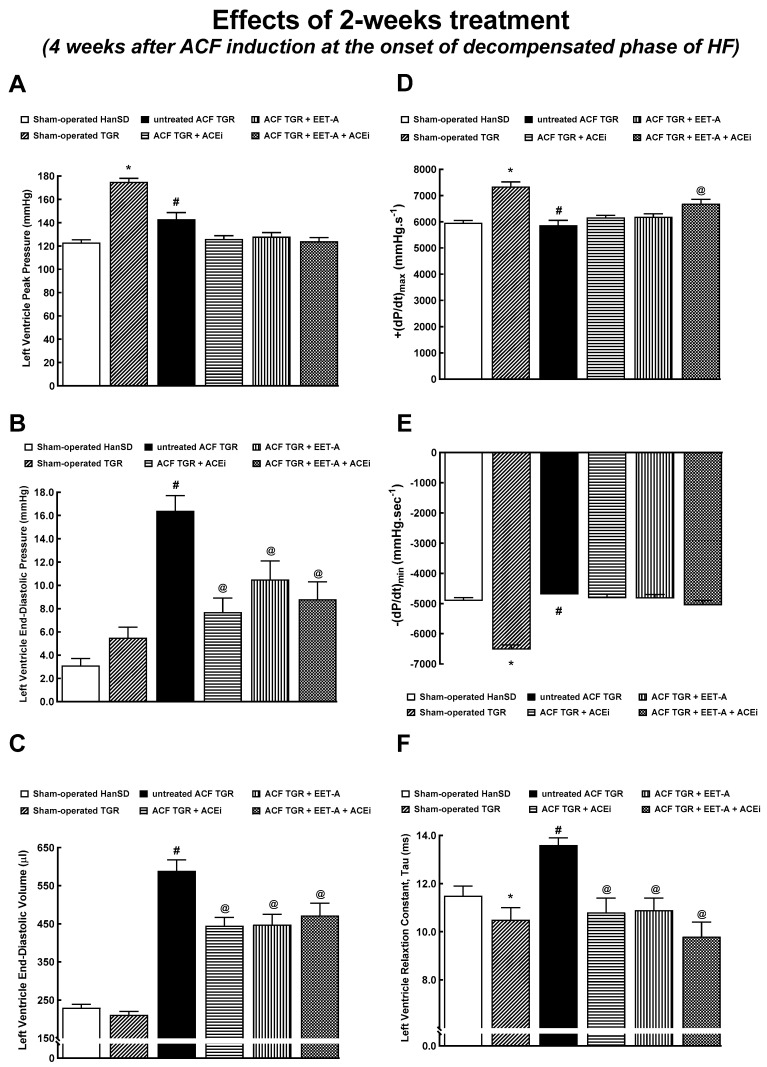
Part 1 of the left ventricular cardiac function assessment by invasive hemodynamic analysis performed 4 weeks after creation of the aorto-caval fistula (ACF) and 2 weeks after initiation of treatment in sham-operated transgene-negative Hannover Sprague-Dawley rats (HanSD), sham-operated heterozygous Ren-2 transgenic rats (TGR) and TGR rats with ACF, untreated or treated with angiotensin-converting enzyme inhibitor (ACEi) alone or with 14,15-epoxyeicosatrienoic acid analog (EET-A) alone or with the combination of EET-A and ACEi. Left ventricle peak pressure (**A**), left ventricle end-diastolic pressure (**B**), left ventricle end-diastolic volume (**C**), maximum rates of pressure rise (+dP/dt)_max_ (**D**), maximum rates of pressure fall (-dP/dt)_max_ (**E**), relaxation constant tau (**F**). * *p* < 0.05 sham-operated TGR versus sham-operated HanSD rats. **^#^** *p* < 0.05 for untreated ACF TGR versus sham-operated TGR. ^@^ *p* < 0.05 versus untreated ACF TGR.

**Figure 5 biomedicines-09-01053-f005:**
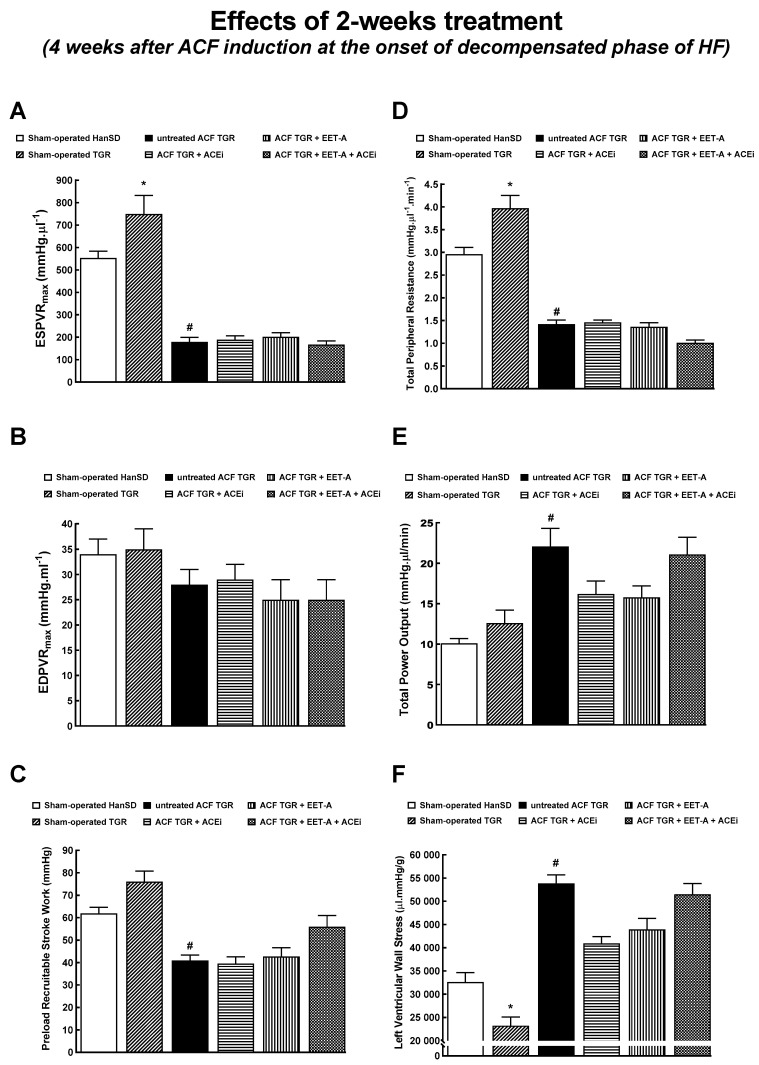
Part 2 of the left ventricular cardiac function assessment by invasive hemodynamic analysis performed 4 weeks after creation of the aorto-caval fistula (ACF) and 2 weeks after initiation of treatment in sham-operated transgene-negative Hannover Sprague-Dawley (HanSD), sham-operated heterozygous Ren-2 transgenic rats (TGR) and TGR rats with ACF, untreated or treated with angiotensin-converting enzyme inhibitor (ACEi) alone or with 14,15-epoxyeicosatrienoic acid analog (EET-A) alone or with combination of EET-A and ACEi. End-systolic pressure-volume relationship (ESPVR) (**A**), end-diastolic pressure-volume relationship (EDPVR) (**B**), preload recruitable stroke work (**C**), total peripheral resistance (**D**), total power output (**E**), left ventricular wall stress (**F**). * *p* < 0.05 for sham-operated TGR versus sham-operated HanSD rats. **^#^** *p* < 0.05 for untreated ACF TGR versus sham-operated TGR.

**Figure 6 biomedicines-09-01053-f006:**
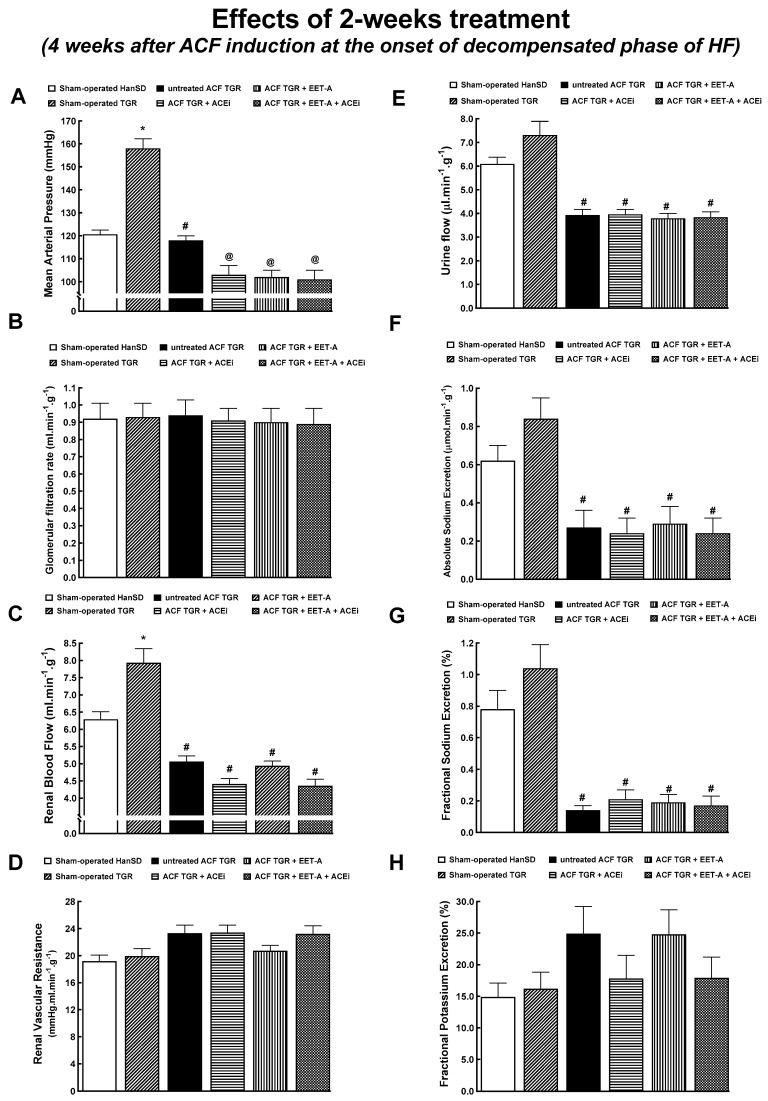
Renal function analysis was performed 4 weeks after the creation of the aorto-caval fistula (ACF) and 2 weeks after initiation of treatment in sham-operated transgene-negative Hannover Sprague-Dawley (HanSD), sham-operated heterozygous Ren-2 transgenic rats (TGR), and TGR rats with ACF, untreated or treated with angiotensin-converting enzyme inhibitor (ACEi) alone or with 14,15-epoxyeicosatrienoic acid analog (EET-A) alone or with the combination of EET-A and ACEi. Mean arterial pressure (**A**), glomerular filtration rate (**B**), renal blood flow (**C**), renal vascular resistance (**D**), urine flow (**E**), absolute (**F**) and fractional sodium excretion (**G**) and fractional potassium excretion (**H**) * *p* < 0.05 for sham-operated TGR versus sham-operated HanSD rats. **^#^** *p* < 0.05 versus sham-operated TGR. ^@^ *p* < 0.05 versus untreated ACF TGR.

**Figure 7 biomedicines-09-01053-f007:**
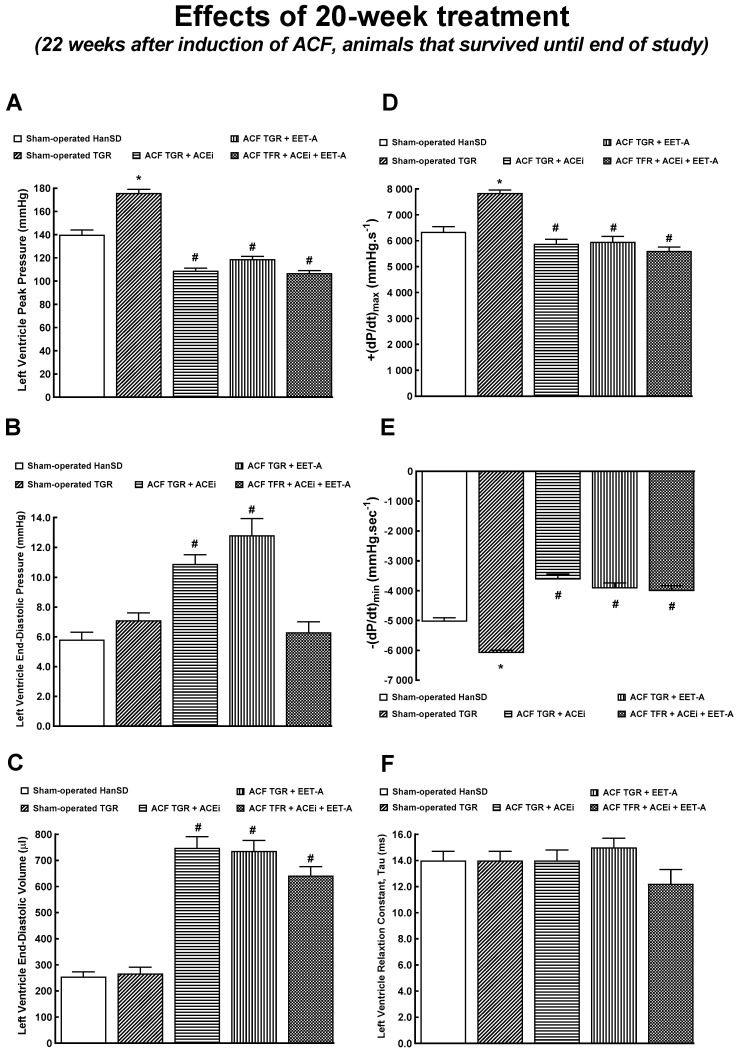
Part 1 of the left ventricular cardiac function assessment by invasive hemodynamic analysis performed 22 weeks after creation of the aorto-caval fistula (ACF) and 20 weeks after initiation of treatment in sham-operated transgene-negative Hannover Sprague-Dawley (HanSD), sham-operated heterozygous Ren-2 transgenic rats (TGR) and TGR rats with ACF treated with angiotensin-converting enzyme inhibitor (ACEi) alone or with 14,15-epoxyeicosatrienoic acid analog (EET-A) alone or with the combination of EET-A and ACEi. Left ventricle peak pressure (**A**), left ventricle end-diastolic pressure (**B**), left ventricle end-diastolic volume (**C**), maximum rates of pressure rise (+dP/dt)_max_ (**D**), maximum rates of pressure fall (-dP/dt)_max_ (**E**), relaxation constant tau (**F**). * *p* < 0.05 for sham-operated TGR versus sham-operated HanSD rats. **^#^** *p* < 0.05 for ACF TGR versus sham-operated TGR.

**Figure 8 biomedicines-09-01053-f008:**
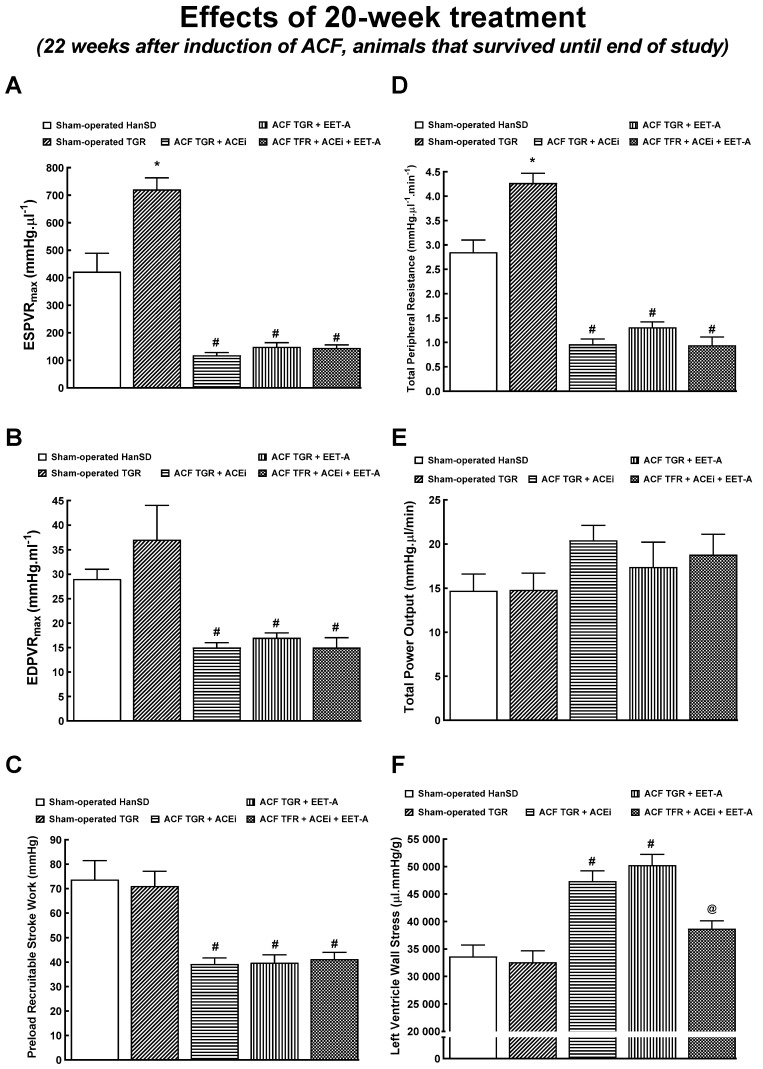
Part 2 of the left ventricular cardiac function assessment by invasive hemodynamic analysis performed 22 weeks after creation of the aorto-caval fistula (ACF) and 20 weeks after initiation of treatment in sham-operated transgene-negative Hannover Sprague-Dawley (HanSD), sham-operated heterozygous Ren-2 transgenic rats (TGR) and TGR rats with ACF, untreated or treated either with angiotensin-converting enzyme inhibitor (ACEi) alone or with 14,15-epoxyeicosatrienoic acid analog (EET-A) alone or with the combination of EET-A and ACEi. End-systolic pressure-volume relationship (ESPVR) (**A**), end-diastolic pressure-volume relationship (EDPVR) (**B**), preload recruitable stroke work (**C**), total peripheral resistance (**D**), total power output (**E**), left ventricular wall stress (**F**). * *p* < 0.05 sham-operated TGR versus sham-operated HanSD rats. **^#^** *p* < 0.05 for ACF TGR versus sham-operated TGR. ^@^ *p* < 0.05 versus other ACF TGR.

**Table 1 biomedicines-09-01053-t001:** Genes analyzed in the kidney cortex.

ID Assay Number	Gene Name	Abbreviation
Rn00561847_m1	renin	Ren
Rn00561094_m1	angiotensin I converting enzyme	Ace
Rn01416293_m1	angiotensin I converting enzyme 2	Ace2
Rn00593114_m1	angiotensinogen (serpin peptidase inhibitor, clade A, member 8)	Agt
Rn02758772_s1	angiotensin II receptor, type 1a	Agtr1a
Rn00562673_s1	MAS1 proto-oncogene, G protein-coupled receptor	Mas1
Hs99999901_s1	18S rRNA ribosomal subunit	18s rRNA
Rn00598510_m1	cytochrome P450, family 4, subfamily a, polypeptide 1	Cyp4a1
Rn01413752_m1	cytochrome P450, family 2, subfamily c, polypeptide 23	Cyp2c23
Rn00567876_m1	adrenoceptor alpha 1A	Adra1a
Rn01471343_m1	adrenoceptor alpha 1B	Adra1b
Rn00562488_s1	adrenoceptor alpha 2A	Adra2a
Rn00593312_s1	adrenoceptor alpha 2B	Adra2b
Rn00593341_s1	adrenoceptor alpha 2C	Adra2c
Rn00824536_s1	adrenoceptor beta 1	Adrb1
Rn00560650_s1	adrenoceptor beta 2, surface	Adrb2
Rn00560677_s1	angiotensin II receptor, type 2	Agtr2

**Table 2 biomedicines-09-01053-t002:** Body and organ weights 4 weeks after creation of the aorto-caval fistula (ACF) or sham-operation and after 2 weeks’ treatment with 14,15-epoxyeicosatrienoic acids analog (EET-A) and angiotensin-converting enzyme inhibitor (ACEi), alone or combined with EET-A.

	Group					
	HanSD	TGR	ACF TGR	ACF TGR	ACF TGR	ACF TGR
	+water	+water	+water	+ACEi	+EET-A	+EET-A + ACEi
Body Weight (g)	465 ± 7	487 ± 9	415 ± 9 ^#^	460 ± 16	453 ± 8	449 ± 7
Tibia length (mm)	38.2 ± 0.3	37.7 ± 0.2	37.5 ± 0.2	37.8 ± 0.2	37.3 ± 0.3	37.4 ± 0.2
Whole heart weight (mg)/tibia length (mm)	37.69 ± 0.74	48.01 ± 0.98 *	68.27 ± 1.07 ^#^	54.49 ± 1.06 ^@^	64.07 ± 1.19	56.69 ± 1.04 ^@^
LV weight (mg)/tibia length (mm)	24.56 ± 0.21	33.69 ± 0.91 *	40.79 ± 0.89 ^#^	33.86 ± 0.24 ^@^	38.07 ± 0.81	35.76 ± 0.26 ^@^
RV weight (mg)/tibia length (mm)	7.28 ± 0.19	7.56 ± 0.21	14.69 ± 0.39 ^#^	11.95 ± 0.18 ^@^	15.26 ± 0.43	11.69 ± 0.21 ^@^
RV weight (mg)/LV weight (mg)	0.296 ± 0.02	0.224 ± 0.01	0.361 ± 0.03 ^#^	0.353 ± 0.02	0.401 ± 0.09	0.327 ± 0.02 ^@^
Lung weight (mg)/tibia length (mg)	50.48 ± 1.06	48.52 ± 1.29	74.89 ± 1.43 ^#^	61.69 ± 1.17 ^@^	68.48 ± 2.39	56.64 ± 1.23 ^@^
Kidney weight (mg)/tibia length (mg)	39.27 ± 0.51	42.19 ± 1.29	40.35 ± 1.13	36.94 ± 1.44	39.04 ± 1.14	37.89 ± 0.71
Liver weight (mg)/tibia length (mg)	455 ± 16	459 ± 17	435 ± 21	449 ± 24	464 ± 26	432 ± 29

Values are the means ± SEM. LV, left ventricle; RV, right ventricle; ***** *p* < 0.05 vs. sham-operated HanSD rats; **^#^** *p* < 0.05 vs. TGR + water; **^@^** *p* < 0.05 vs. ACF TGR + water.

**Table 3 biomedicines-09-01053-t003:** Echocardiographic analysis 4 weeks after the induction of aorto-caval fistula (ACF) or sham-operation and after 2 weeks’ treatment with 14,15-epoxyeicosatrienoic acids analog (EET-A) and angiotensin-converting enzyme inhibitor (ACEi) alone or combined with EET-A.

	Group					
	HanSD	TGR	ACF TGR	ACF TGR	ACF TGR	ACF TGR
	+water	+water	+water	+ACEi	+EET-A	+EET-A + ACEi
Heart rate (min^−1^)	371 ± 11	376 ± 10	356 ± 9	368 ± 8	364 ± 8	354 ± 11
LV diastolic diameter (mm)	6.64 ± 0.13	6.14 ± 0.16	9.99 ± 0.24 ^#^	8.95 ± 0.21	9.83 ± 0.23	10.27 ± 0.25
LV systolic diameter (mm)	3.39 ± 0.16	2.99 ± 0.11	6.06 ± 0.27 ^#^	5.36 ± 0.21	6.01 ± 0.19	6.37 ± 0.21
LV anterior wall thickness in diastole (mm)	2.06 ± 0.06	2.76 ± 0.04 *	2.18 ± 0.05 ^#^	2.08 ± 0.04	2.06 ± 0.05	1.68 ± 0.02 ^@^
LV posterior wall thickness in diastole (mm)	2.22 ± 0.06	3.11 ± 0.08 *	2.47 ± 0.08 ^#^	2.39 ± 0.06	2.36 ± 0.06	1.87 ± 0.02 ^@^
LV relative wall thickness	0.67 ± 0.02	1.02 ± 0.06 *	0.48 ± 0.02 ^#^	0.54 ± 0.03	0.48 ± 0.02	0.37 ± 0.02 ^@^
LV ejection fraction (%)	78.2 ± 1.7	81.6 ± 0.9	67.2 ± 1.1 ^#^	66.8 ± 1.3	68.4 ± 1.4	65.2 ± 1.2
LV fractional shortening (%)	48.5 ± 1.7	51.3 ± 0.8	39.6 ± 1.1 ^#^	40.3 ± 1.1	38.9 ± 1.1	37.6 ± 0.9
LV stroke volume (µL)	179 ± 6.4	156 ± 9.9	379 ± 19.3 ^#^	303 ± 11	364 ± 18.4	376 ± 18.2
Cardiac output (mL/min)	60.9 ± 1.7	58.5 ± 3.6	133.6 ± 5.5 ^#^	111.5 ± 4.6	132.8 ± 6.8	139.5 ± 6.1
RV basal diameter in diastole (mm)	3.23 ± 0.08	3.19 ± 0.11	5.61 ± 0.36 ^#^	4.26 ± 0.26	4.79 ± 0.25	5.62 ± 0.18
RV midcavity diameter in diastole (mm)	3.07 ± 0.09	2.67 ± 0.14	5.01 ± 0.43 ^#^	3.88 ± 0.24	4.05 ± 0.20	5.28 ± 0.16
RV fractional area change (%)	54.7 ± 1.9	55.7 ± 2.8	50.3 ± 2.9	49.4 ± 2.1	51.5 ± 3.2	44.1 ± 2.9

Values are the means ± SEM. LV, left ventricle; RV, right ventricle; ***** *p* < 0.05 vs. sham-operated HanSD rats; **^#^** *p* < 0.05 vs. TGR + water; **^@^** *p* < 0.05 vs. ACF TGR + water.

**Table 4 biomedicines-09-01053-t004:** Body and organ weights 22 weeks after creation of the aorto-caval fistula (ACF) or sham-operation and after 20-weeks’ treatment with 14,15-epoxyeicosatrienoic acids analog (EET-A) and angiotensin-converting enzyme inhibitor (ACEi), alone or combined with EET-A.

	Group				
	HanSD	TGR	ACF TGR	ACF TGR	ACF TGR
	+water	+water	+ACEi	+EET-A	+EET-A + ACEi
Body Weight (g)	630 ± 10	633 ± 9	689 ± 8 ^#^	701 ± 19	608 ± 18 ^@^
Tibia length (mm)	43.8 ± 0.2	43.7 ± 0.3	44.1 ± 0.4	44.2 ± 0.4	44.1 ± 0.4
Whole heart weight (mg)/tibia length (mm)	33.67 ± 0.58	46.69 ± 0.79 *	65.09 ± 1.12 ^#^	66.12 ± 1.22 ^#^	55.01 ± 1.09 ^@^
LV weight (mg)/tibia length (mm)	26.94 ± 0.23	34.55 ± 0.89 *	39.22 ± 0.27 ^#^	40.27 ± 0.29 ^#^	33.33 ± 0.21 ^@^
RV weight (mg)/tibia length (mm)	7.47 ± 0.21	7.57 ± 0.18	15.87 ± 0.35 ^#^	15.77 ± 0.33 ^#^	11.36 ± 0.22 ^@^
RV weight (mg)/LV weight (mg)	0.277 ± 0.03	0.219 ± 0.03	0.405 ± 0.04 ^#^	0.392 ± 0.03 ^#^	0.341 ± 0.03 ^@^
Lung weight (mg)/tibia length (mg)	49.77 ± 1.18	48.06 ±1.23	61.91 ± 1.51 ^#^	66.97 ± 1.78 ^#^	51.29 ± 0.98 ^@^
Kidney weight (mg)/tibia length (mg)	42.69 ± 0.99	42.56 ± 1.17	39.01 ± 1.23	40.27 ± 1.16	39.23 ± 1.25
Liver weight (mg)/tibia length (mg)	458 ± 19	440 ± 23	444 ± 27	437 ± 25	447 ± 28

Values are the means ± SEM. LV, left ventricle; RV, right ventricle; * *p* < 0.05 vs. sham-operated HanSD rats; **^#^** *p* < 0.05 vs. TGR + water; **^@^** *p* < 0.05 vs. ACF TGR + ACEi and vs. ACF TGR + EET-A.

**Table 5 biomedicines-09-01053-t005:** Echocardiographic analysis 22 weeks after the induction of aorto-caval fistula (ACF) or sham-operation and after 20 weeks´ treatment with 14,15-epoxyeicosatrienoic acids analog (EET-A) and angiotensin-converting enzyme inhibitor (ACEi) alone or combined with EET-A.

	Group				
	HanSD	TGR	ACF TGR	ACF TGR	ACF TGR
	+water	+water	+ACEi	+EET-A	+EET-A + ACEi
Heart rate (min^−1^)	372 ± 9	365 ± 18	363 ± 9	365 ± 11	368 ± 19
LV diastolic diameter (mm)	6.96 ± 0.22	7.09 ± 0.27	11.29 ± 0.31 ^#^	11.12 ± 0.51 ^#^	11.51 ± 0.29 ^#^
LV systolic diameter (mm)	3.78 ± 0.22	4.35 ± 0.39	7.55 ± 0.26 ^#^	7.63 ± 0.44 ^#^	7.69 ± 0.37 ^#^
LV anterior wall thickness in diastole (mm)	2.32 ± 0.06	3.05 ± 0.06 *	2.06 ± 0.05 ^#^	2.25 ± 0.06 ^#^	1.87 ± 0.05 ^#^
LV posterior wall thickness in diastole (mm)	2.55 ± 0.06	3.07 ± 0.11 *	2.08 ± 0.05 ^#^	2.12 ± 0.07 ^#^	2.02 ± 0.05 ^#^
LV relative wall thickness	0.738 ± 0.04	0.913 ± 0.04 *	0.368 ± 0.02 ^#^	0.393 ± 0.02 ^#^	0.369 ± 0.03 ^#^
LV ejection fraction (%)	75.8 ± 1.9	71.6 ± 2.6	55.8 ± 1.6 ^#^	53.9 ± 1.4 ^#^	57.3 ± 1.5 ^#^
LV fractional shortening (%)	46.1 ± 1.7	44.6 ± 1.6	33.3 ± 0.7 ^#^	32.5 ± 1.1 ^#^	32.4 ± 0.8 ^#^
LV stroke volume (µL)	191 ± 11	174 ± 8	437 ± 22 ^#^	408 ± 29 ^#^	414 ± 31 ^#^
Cardiac output (mL/min)	69.9 ± 1.5	63.6 ± 4.2	158 ± 6.2 ^#^	152 ± 10.1 ^#^	155 ± 8.7 ^#^
RV basal diameter in diastole (mm)	3.06 ± 0.13	3.08 ± 0.11	6.65 ± 0.19 ^#^	5.81 ± 0.36 ^#^	5.41 ± 0.13 ^@^
RV midcavity diameter in diastole (mm)	2.45 ± 0.11	2.49 ± 0.12	6.03 ± 0.16 ^#^	5.28 ± 0.21 ^#^	5.02 ± 0.12 ^@^
RV fractional area change (%)	61.9 ± 5.8	65.1 ± 5.2	35.9 ± 1.1 ^#^	39.5 ± 1.6 ^#^	33.7 ± 1.4 ^#^

Values are means ± SEM. LV, left ventricle; RV, right ventricle; * *p* < 0.05 vs. sham-operated HanSD rats; **^#^** *p* < 0.05 vs. TGR + water; **^@^** *p* < 0.05 vs. ACF TGR + ACEi.

## Data Availability

Data is contained within the article and Appendix A.

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
