# Peer review of "Effects of Epoxyeicosatrienoic Acid-Enhancing Therapy on the Course of Congestive Heart Failure in Angiotensin II-Dependent Rat Hypertension: From mRNA Analysis towards Functional In Vivo Evaluation"

_biomedicines, 2021, doi:10.3390/biomedicines9081053_

Round 1

Reviewer 1 Report

This manuscript reports the combination of EET analog, EET-A, and ACEi may have beneficial effect to the heart failure induced by aorto-caval fistula in Ren-2 transgenic Rat, a model of hypertension and increased RAS activity. Treatment improves its survival rate, however the mechanism of prolonged survival stays unclear. Neither renal function nor cardiac function improved significantly. It certainly suggests the new approach for the treatment of HF, however there are several concerns.

First of all, although it is not a pure scientific concern, the manuscript is written in a way for the readers to get easily lost. Especially the results section is too long. The authors should summarize the results better rather than describing all figures and tables one by one.

The survival rate was significantly prolonged by the treatment, however the mechanism is unclear. Either renal or cardiac function have not been significantly improved to explain the beneficial effects of the treatment.

All bar graphs will be more visually improved if they use dot plot. For example, in figure 1A, it is hard to believe that there are no statistically significant differences between groups.

It is not clear which p value was obtained by student-t test or ANOVA. All analysis presented in the manuscript need ANOVA since they compare among multiple variants.

Titles of figures 1-6 are the same, so as 8-10, 11-12.

In Figure 4, there is no description of the blotting images. I assume they are Western blot results, but then no identification of the each lane. Figure 4H indicates significant increase of sEH/GAPDH in ACF TGR group, but since loading control intensity varies, it may due to the decreased level of GAPDH.

In Figure 6B, why albuminuria is worse in sham-operated TGR compared to untreated ACF TGR? At 20weeks, is sham-operated HanSD abnormal, or all treated groups are significantly decreased compared to the “normal”? If the authors can provide HanSD data at 20weeks without any surgery, it might be informative.

Table 2& 3 show thinner wall thickness in combined treatment group than in controls, increased LV diastolic diameter and reduced cardiac function. Does this mean the combined treatment cause dilated cardiomyopathy rather than treating heart failure 2 weeks after the treatment initiated and continued to 22 weeks?

Figure 10 concludes there are no effects of any treatment in renal function compared to no treatment. I agree that it may be due to early evaluation time point and effects have not been established as the authors explained in the discussion (ln 740-745). It will be better if they clarify why they use 2 weeks-time-point rather than 4 weeks in section3.5.

In discussion, ln 863-865, where is the three weeks echo data?

There are 4 tables in supplement, but they seem to be same tables in the main body.

Author Response

We appreciate the reviewers´ comments on our study, and we are grateful for their encouraging and constructive suggestions.

Responses to reviewer #1:

Reviewers Comment: …….. First of all, although it is not a pure scientific concern, the manuscript is written in a way for the readers to get easily lost. Especially the results section is too long. The authors should summarize the results better rather than describing all figures and tables one by one.

Authors Response:

We agree that the result section is too long and somewhat difficult to follow for readers, hence we have simplified it in the revised manuscript. Specifically, the results from Series 1 and 2 are now presented together (new Section 3.1). These two series quantify the degree of activation of the vasoactive/neurohormonal systems in the very early phase of the ACF-induced HF in ACF TGR (i.e. before the treatment was initiated) and it is not the primary message of our present study. We have decided to present these data in Supplemental Figures 1 to 6, and for the main body of the manuscript, we have prepared a new Figure 1. This new figure summarizes the most important finding of these series, which is that two weeks after the induction of ACF in TGR exhibited markedly decreased tissue availability of biologically active epoxygenase products. We have also modified the appropriate Discussion section (4.1. “Activity of the neurohormonal systems…”), again in order to discuss only the most relevant results. The same is valid for all other Results section, in attempt to emphasize only the most relevant findings rather than to describe separately every panel in the figures.

In addition, we have prepared new Figures 2 and 3 (converted from the original Figure 7), so now the effects on the survival rate and albuminuria are presented in the individual figures. The strategic aim of these changes is to make our presentation more straightforward for readers and present the most important findings immediately at the beginning of the Results section. All these changes thorough the Results and Discussion sections in the revised manuscript are shown in red font.

Reviewers Comment: …….. The survival rate was significantly prolonged by the treatment, however, the mechanism is unclear. Either renal or cardiac function has not been significantly improved to explain the beneficial effects of the treatment.

Authors Response:

We can only say that we agree and we are glad that the main message of our study has been appreciated. There is no doubt that all treatment regimens improves the survival rate, but the beneficial effects are not simply mediated either via renal or cardiac mechanisms. We discuss this issue openly and in relative detail in the Discussion section, specifically in the Discussion section 4.3. entitled “Mechanisms of the beneficial actions of the treatments”. We added to this section more detailed explanation regarding the effects on the renal function (page 24, line 745 to 755 and page 25, line 792 to 797, all shown in red font). Our present findings confirm that the observed beneficial action cannot be simply mediated by a single improvement of one parameter, but it is rather a synergistic combination of minute, positive effects on manifold parameters (systems, functions). Nevertheless, it is obvious that future studies are needed to address this issue and it is openly mentioned in the Discussion section of our manuscript.

Reviewers Comment: …….. All bar graphs will be more visually improved if they use dot plot. For example, in figure 1A, it is hard to believe that there are no statistically significant differences between groups.

Authors Response:

We fully agree and therefore we prepared new figures. It was also secondary reason why we have decided to present survival rate and albuminuria in separate figures. In addition for the presentation of mRNA expression data in the kidney, we prepared scatter plot bar diagrams and we believe that it is now clear (Supplemental figure 1). Additionally, once again we have tested the statistical significance of changes between groups and we have confirmed previous findings. It is important to recognize that the values in the graphs of mRNA analysis represent log2 n-fold gene expression and the changes in experimental groups were relatively small and insignificant (with an exception of the renin gene). Moreover, we did not find any scientific (statistical) reasons to exclude obtained data, especially considering the sizable biological variability, which is very common in such studies. However, we agree that our original bar graphs were not entirely perspicuous, therefore we modified them as suggested.

In addition, we have put more emphasis on the fact that the regulation of all vasoactive/neurohormonal systems is an extremely complex issue, therefore sole analysis of mRNA and/or protein level can be sometimes misleading and to make solid conclusions more comprehensive analysis is required. We mention this issue in the revised version of the Discussion section (4.1. “Activity of the neurohormonal systems…”; page 22, line 651 to 655, shown in red font).

Reviewers Comment: …….. It is not clear which p value was obtained by student-t test or ANOVA. All analysis presented in the manuscript need ANOVA since they compare among multiple variants.

Authors Response:

We apologize for this error, certainly all statistical analysis (with an exception of the effects on the survival rate) were done by one-way ANOVA. This mistake was corrected (page 6, line 252 to 253, shown in red font).

Reviewers Comment: …….. Titles of figures 1-6 are the same, so as 8-10, 11-12.

Authors Response:

We have recognized that it was another drawback of our original figures and therefore in new figures we have modified the titles so they are more explicit.

Reviewers Comment: …….. In Figure 4, there is no description of the blotting images. I assume they are Western blot results, but then no identification of the each lane. Figure 4H indicates significant increase of sEH/GAPDH in ACF TGR group, but since loading control intensity varies, it may due to the decreased level of GAPDH.

Authors Response:

The western blot images visible on graphs are only representative blots of each analyzed protein, which is a requirement of the editorial office. The expression of each protein was calculated based on at least 6 – 8 samples (usually more) in each group. We are aware that the variability of the control protein GAPDH might contribute to the “false” increase in the expression of the analyzed protein, but we have carefully reviewed our results and we believe that the observed increase is dominantly mediated by a real change in sEH expression. We have sent the original images of the gels (with description of groups) to the editorial office as a supplemental file. However, after careful consideration of the reviewer’s suggestions and since it is not the most important finding of our study, but rather supportive data, in the revised version of the manuscript these results are now presented in the appropriate form in the Supplemental Figure 4.

Reviewers Comment: …….. In Figure 6B, why albuminuria is worse in sham-operated TGR compared to untreated ACF TGR? At 20weeks, is sham-operated HanSD abnormal, or all treated groups are significantly decreased compared to the “normal”? If the authors can provide HanSD data at 20weeks without any surgery, it might be informative.

Authors Response:

We are very pleased that this surprising finding was acknowledged. We have not addressed it in the original manuscript, because even if it is of special interest it is not the main message of our study and we did not want to disturb the readers from following our main hypothesis. After careful consideration we have decided to discuss it in the revised version of our manuscript together with the “age-dependent” increase in albuminuria in sham-operated HanSD rats. We have placed this new discussion purposely in the Results section (“Section 3.2. Series 3: Effects of single treatment with EET-A….”) in order not to distract the readers from the main topic, but to make it easily available for all particularly interested in this phenomenon.   

Reviewers Comment: …….. Table 2& 3 show thinner wall thickness in combined treatment group than in controls, increased LV diastolic diameter and reduced cardiac function. Does this mean the combined treatment cause dilated cardiomyopathy rather than treating heart failure 2 weeks after the treatment initiated and continued to 22 weeks?

Authors Response:

We highly appreciate that this was noted. We agree that these findings indicate (at least from our point of view) that even if the combined treatment improved the survival rate, it is not the optimal treatment approach for this model of high-output HF, in which the progressive development of dilated cardiomyopathy is one of the critical hallmarks. We address this issue in the Discussion section of the revised manuscript (Discussion section 4.3. “Mechanisms of the beneficial actions of the treatments”; page 25, line 792 to 797, shown in red font).

Reviewers Comment: …….. Figure 10 concludes there are no effects of any treatment in renal function compared to no treatment. I agree that it may be due to early evaluation time point and effects have not been established as the authors explained in the discussion (ln 740-745). It will be better if they clarify why they use 2 weeks-time-point rather than 4 weeks in section3.5.

Authors Response:

We agree and we did as suggested (Result section 3.4; page 16, line 485 to 497, shown in red font). We also prepared a new Figure 6 in which data regarding renal function is presented in more straightforward way for the readers. Finally, we address and clarify this issue in more detail in the Discussion (section 4.3. entitled “Mechanisms of the beneficial actions of the treatments”; page 24, line 745 to 755, shown in red font).

Reviewers Comment: …….. In discussion, ln 863-865, where is the three weeks echo data?

Authors Response:

We apologize for this obvious mistake, it should be “four weeks”. It was corrected (page 25, line 765, shown in red font).

Reviewers Comment: …….. There are 4 tables in supplement, but they seem to be same tables in the main body.

Authors Response:

We apologize for this mistake, all tables were deleted from the supplemental material.

Reviewer 2 Report

Kala et. al present a series of animal, qPCR, biochemical and pharmacological studies in order to investigate the effects of epoxyeicosatrienoic acid-enhancing therapy on the course of congestive heart failure in angiotensin II-dependent rat hypertension. It is very large study, using several techniques. The measurements and the analysis provided data and the authors made right conclusions. Generally speaking, I really appreciate the huge work devoted this work, and I do not have major criticisms to make. Although, I would like to suggest few things as listed below:

The abbreviation section is located in the Material and Methods chapter, which is a strange place for it, at least to my taste. The letters in the figures (i.e. below the columns) are very tiny. Please, increase the size significantly.

Line 924 and 945: I was not able to find any supplementary material in the manuscript. So, either delete these rows or you may want to make some supplementary figures (i.e. Figure 2 and 4 may be converted to Supplementary figures).

Author Response

We appreciate the reviewers´ comments on our study, and we are grateful for their encouraging and constructive suggestions.

Responses to reviewer #2:

Reviewers Comment: …….. The abbreviation section is located in the Material and Methods chapter, which is a strange place for it, at least to my taste. The letters in the figures (i.e. below the columns) are very tiny. Please, increase the size significantly.

Authors Response:

We are grateful for this comment, since the clarity of the graphs is crucial for the full understanding of our complex study. It was corrected as suggested and also we moved the Abbreviations to the correct place i.e. before the References section (from page 26, line 855 to page 27, line 897, shown in red font).

Reviewers Comment: …….. Line 924 and 945: I was not able to find any supplementary material in the manuscript. So, either delete these rows or you may want to make some supplementary figures (i.e. Figures 2 and 4 may be converted to Supplementary figures).

Authors Response:

Thank You for this comment, of course in the original version of the manuscript it was an obvious mistake. However, in the revised version of the manuscript we have prepared new Supplementaty material. According to the reviewers’ comments, in order to make the manuscript more comprehensive, we have decided to present only the most important findings in the main body of the manuscript; therefore some of the material from Series 1 and 2 was moved to the Supplement. We believe that this modification (i.e. reducing the number of Figures in the main body of the manuscript) will help the readers to follow our complex study.